# Advances in RNA-Silencing-Related Resistance against Viruses in Potato

**DOI:** 10.3390/genes13050731

**Published:** 2022-04-22

**Authors:** Lili Jiang, Zunhe Du, Guizhi Zhang, Teng Wang, Guanghui Jin

**Affiliations:** College of Agriculture, Heilongjiang Bayi Agricultural University, 5 Xinfeng Road, Development Zone, Daqing 163319, China; dzh_115@163.com (Z.D.); damao98@163.com (G.Z.); wteng1129@126.com (T.W.); ghjin1122@163.com (G.J.)

**Keywords:** RNA silencing, siRNA, miRNA, amiRNA, potato virus

## Abstract

Potato is a major food crop that has the potential to feed the increasing global population. Potato is the fourth most important crop and a staple food for many people worldwide. The traditional breeding of potato poses many challenges because of its autotetraploid nature and its tendency toward inbreeding depression. Moreover, potato crops suffer considerable production losses because of infections caused by plant viruses. In this context, RNA silencing technology has been successfully applied in model and crop species. In this review, we describe the RNA interference (RNAi) mechanisms, including small-interfering RNA, microRNA, and artificial microRNA, which may be used to engineer resistance against potato viruses. We also explore the latest advances in the development of antiviral strategies to enhance resistance against potato virus X, potato virus Y, potato virus A, potato leafroll virus, and potato spindle tuber viroid. Furthermore, the challenges in RNAi that need to be overcome are described in this review. Altogether, this report would be insightful for the researchers attempting to understand the RNAi-mediated resistance against viruses in potato.

## 1. Introduction

Potato (*Solanum tuberosum* L.) is ranked fourth worldwide among all food crops in terms of its total production, next to rice, wheat, and corn. Potato continues to be recommended as an important component of diversified cropping systems, especially in the face of the current population expansion, to overcome problems in food supply, nutrition, and food security as well as global climate change challenges [1,2]. In 2020, 16.5 million hectares of potato crops were cultivated worldwide, which yielded 217.7 million tons of potatoes [3].

Small-interfering RNAs (siRNAs) were first described by Hamilton and Baulcombe in 1999, who confirmed that naturally occurring siRNAs in plants can cause post-transcriptional gene silencing [4]. Plant viruses could pose serious threats to a wide range of crops. More specifically, plant viruses are endemic throughout the potato-growing areas and significantly reduce the yield of various potato varieties [5]. It has been reported that approximately 40 different viral species and two viroids could infect potato crops, with the viral impact being enhanced in subsequent generations [6].

RNA silencing (also termed RNA interference, RNAi) provides an antiviral immune system that can be applied to plants and has attracted increasing attention in plant–virus interaction studies [7,8,9]. RNA silencing is the best-studied antiviral defense mechanism in plants [10]. RNA silencing is a sequence-specific regulatory mechanism that is triggered by double-stranded RNA (dsRNA) in eukaryotic organisms and could provide resistance against the invasion of exogenous nucleic acids via post-transcriptional gene silencing (PTGS) or transcriptional gene silencing (TGS) [11]. TGS occurs in the cell nucleus, with the cytoplasm as the target, and the methylation of gene promoters and histones causes the inactivation of genes with transcriptional activity [12]. PTGS uses the mRNA produced by gene transcription as the target, and the promoter of the gene is transcribed to mRNA; however, the mRNA is degraded and is untranslatable [13].

In the RNA silencing pathway, dsRNA is cleaved into fragments of 18–27 nucleotides and non-coding interfering RNAs, such as siRNAs and microRNAs (miRNAs). Both siRNAs and miRNAs may be associated with various plant proteins to form the RNA-induced silencing complex (RISC) [14,15,16].

In RNA-containing plant viruses, the dsRNA structures of viral RNAs are processed by Dicer-like proteins into long (24–26 nt) and short (21–23 nt) siRNA fragments [17,18]. Double-stranded siRNAs are unwound into the sense (passenger) and antisense (guide) strands by the activated RISC. Subsequently, the passenger strand is degraded, whereas the guide strand is incorporated into the functional siRISC, which contains an argonaute (AGO) protein encompassing an sRNA domain (PAZ domain). Then, the guide strand targets a complementary mRNA transcript via a base-pairing interaction [19,20,21]. These viral RNA-cleavage products can be amplified by endogenous plant RNA-dependent RNA (RdRP) polymerases, which gradually inhibit viral protein biosynthesis and strengthens the silencing response (Figure 1a) [22].

miRNAs are a class of short non-coding RNAs that contain approximately 20–24 nucleotides and are derived from endogenous precursor transcripts containing double-stranded (typically hairpin) regions in plants [23,24]. Primary miRNAs (pri-RNAs), which are produced from *MIR* genes by RNA polymerase II (Pol II), have a hairpin-like structure consisting of a terminal loop, an upper stem, the miRNA/miRNA* region (miRNA* refers to the passenger strand), a lower stem, and two arms [25]. Short precursor-miRNAs are produced in the nucleus via the cleavage of pri-miRNAs by DCL1 with the help of HYPONASTIC LEAVES 1 (HYL1) [26]. Pre-miRNAs are further processed by DCL1 into duplex miRNAs. Duplex miRNAs are methylated by the sRNA-specific methyltransferase HUA ENHANCER 1 (HEN1) into mature miRNA duplexes and then transported to the cytoplasm [27]. The guide strand of mature miRNAs associates with the RISC containing AGO, which possesses a catalytic domain (PAZ domain), and the guide RISC to target mRNAs and enhance the potentiation of RNAi (Figure 1b) [28].

Artificial microRNAs (amiRNAs) are a class of artificial small RNAs that have been engineered to silence specific transcripts in plants. amiRNAs consist of two components: a pri-miRNA scaffold and an siRNA insert. This pre-amiRNA is processed via a series of DCL1 slicing events to generate mature amiRNA/amiRNA* duplexes [29]. The mature miRNA sequences within pri-miRNAs are replaced by customized 21-nt RNA fragments that are complementary to the viral targets and show features that are favorable for AGO loading [30]. When entering plant cells, pri-amiRNAs should be transcribed and processed into mature miRNAs with the designed specificity by the host miRNA biogenesis machinery to confer virus resistance (Figure 1c) [18].

In this review, we have focused on siRNA- and miRNA-based strategies for the engineering of virus resistance in potatoes, and we present an overview of the RNA-silencing-mediated control of plant viral disease in potato, e.g., potato virus X (PVX), potato virus Y (PVY), potato virus A (PVA), potato leafroll virus (PLRV), and potato spindle tuber viroid (PSTVd). Because of the large number of publications on RNA silencing, a summary of this topic is provided in Table 1.

## 2. RNA Silencing-Mediated Potato Virus X Resistance

PVX is a type member of the genus *Potexvirus* and a positive-strand RNA virus that infects potatoes [42,43]. Although it normally causes mild symptoms, a severe mosaic disease sometimes occurs, with tuber yield losses in the range of 5%–20% [44,45]. PVX resistance can be achieved via the expression of viral suppressors of RNA silencing. The silencing suppressor P25 protein encoded by PVX is a multifunctional protein with nucleotide binding that act as well as a cell-to-cell movement protein [46,47]. Moreover, *Nicotiana benthamiana* plants transgenic for P25 are less susceptible to PVX [48].

The development of PVX-based virus-induced gene silencing (VIGS) vectors in transgenic plants was central to the suppression of silencing; furthermore, VIGS triggers an RNA-mediated defense mechanism that directly targets the integrity of the invading viral genome [49]. The PVX-based VIGS vector (PVX.PDS_AS_) constructed by Faivre et al. [31] was demonstrated to have PVX resistance in both wild diploid and cultivated tetraploid *Solanum* species. This silencing state can be transmitted and detected for several generations via vegetative propagation.

Another study has reported that virus X-based miRNA silencing (VbMS) vectors silence endogenous miRNAs in potato. The researchers designed the target mimic molecules of miRNA and demonstrated that the VbMS vectors expressed in potato (cv. Katahdin and Russet Burbank) by *Agrobacterium* infiltration bind directly to the endogenous miRNA and block its function [32].

## 3. RNA Silencing-Mediated Potato Virus Y Resistance

PVY is an aphid-borne virus of the genus *Potyvirus* belonging to the family *Potyviridae* and is one of the most important plant viruses affecting potato production [50]. PVY spreads easily and depresses yields by up to 80%. The primary symptoms of PVY are necrosis, mottling, yellowing of leaflets, leaf dropping, and sometimes premature death [51]. Control of PVY infection is difficult in potato because it is propagated vegetatively, which renders the primary viral infection more destructive and persistent across generations. Improvement of PVY viral resistance relies mainly on the expression of virus-derived sense or antisense sequences. The potyviral helper-component protease (HC-Pro) is a multifunctional protein that participates in aphid transmission and can increase viral pathogenicity via the suppression of PTGS in the host plant [52,53]. Petrov et al. [33] proposed the RNAi-based vaccination of potato seedlings with specific viral HC-Pro gene siRNAs, and PTGS was induced in potato plants (cv. Agria). The viral replication and its systemic spread were effectively blocked in the newly grown leaves. Missiou et al. [34] expressed a dsRNA derived from the 3′ terminal of the coat protein gene of PVY in transgenic potatoes of the variety “Spunta,” with the results showing that 80% of the transgenic lines produced siRNAs and were highly resistant to three strains of PVY (PVY^N^, PVY^O^, and PVY^NTN^).

The combination of transgene and transcription strength is functional for constructing efficient resistance to viruses. The data generated from high-throughput sequencing (HTS) can be analyzed to identify the regulatory micro RNAs (miRNA) that are involved in gene regulation as well as the small inhibitory RNAs (siRNA) that confer virus resistance. Potato cv. Atlantic and Ranger Russet were transformed with an inverted hairpin RNA (ihRNA)-generating construct, with the tail-to-tail repeat of the PVY coat protein (PVY-CP)-coding region set apart, and HTS data indicated high levels of siRNA production from the transgenic hairpin construct [35].

## 4. RNA Silencing-Mediated Potato Virus A Resistance

PVA is a member of the genus *Potyvirus* within the picorna-like supergroup of single-stranded, positive-sense RNA viruses [54,55]. Although PVA is quite similar to PVY in structure and behavior, PVA can cause severe mosaic symptoms and damage to potato. The antiviral defense pathways based on RNA silencing can also be utilized for the targeted systemic suppression of host-gene expression [56]. The viral genome-linked (VPg) protein plays a central role in several stages of potyviral infection [36,55]. VPg is a well-known virulence factor in PVA, and the expression of PVA-VPg could suppress the onset and signaling of transgene-mediated RNA silencing and the accumulation of siRNAs. However, VPg acts as a suppressor of RNA silencing, which suggests that its interaction with SGS3 may be important, especially for the suppression of sense-mediated RNA silencing [57].

The PVA-CP has an intrinsic capacity to self-assemble into filamentous virus-like particles. However, the mechanism responsible for the initiation of viral RNA encapsidation in vivo remains unclear. In addition to virion assembly, PVA-CP is also involved in the inhibition of viral RNA translation [55].

Chuang et al. [37] documented that 200-bp segments derived from the coding sequences of PVA were expressed as inverted repeat double-stranded RNAs to generate transgenic potato plants and that the transgenic potato lines tested showed 100% resistance to infection by PVA.

## 5. RNA Silencing-Mediated Potato Leafroll Virus Resistance

PLRV, which is a member of the genus *Polerovirus* within the *Luteoviridae* family, causes a yield loss of >20 million tons of infected potatoes every year [58,59,60]. Inducing RNA silencing against viral RNA or expressing viral RNAs that disturb the viral infection cycle is one of most effective strategies to build plant PLRV resistance. The movement protein (MP) plays a crucial role in PLRV transmission because it helps the cell-to-cell transmission of mature virions or sometimes naked viral genomes [61]. Kumari et al. [38] designed siRNA constructs against the MP of PLRV, and the agroinfiltrated plants did not show PLRV infection. The suppression of viral infection could be attributed to the reduced expression of the MP because of its silencing by the siRNA constructs.

Recently, Kumar et al. used the RNAi approach to prevent PLRV genome packaging and assembly via the suppression of *CP* gene expression in potato. Potato plants that were agroinfiltrated with siRNA constructs against a CP containing an ATPase domain exhibited no rolling symptoms upon PLRV infection. This finding further confirmed that the silencing of CP gene expression is an efficient method for generating PLRV-resistant potato plants [39]. In addition to the *CP* and *MP* genes, ORF2b of PLRV could also confer resistance to PLRV via a PTGS mechanism, and grafting infection experiments showed that resistant transgenic plants could be obtained in this way [58].

Orbegozo et al. [40] reported that a fragment of 395 bp from the coat protein encoding a sequence from PLRV was constructed into a Cre-loxP system of the nptII selectable maker gene to obtain a marker-free transgenic potato. The transgenic lines exhibited extremely high levels of resistance to PLRV, and the resistance was also maintained over tuber generations.

## 6. RNA Silencing-Mediated Potato Spindle Tuber Viroid Resistance

PSTVd is a member of the viroid family *Pospiviroidae*. PSTVd alone or in combination with other viruses can decrease the yield of susceptible potato varieties from 40% to 70% [62]. Symptomatic potatoes show severe leaf curling and stunting; the tubers become reduced in size, with an elongated or spindle-shaped morphology, in response to a severe isolate of PSTVd [63].

RNAi-based strategies have been successfully applied for the development of plants that are resistant to PSTVd. Bao et al. reported that the silencing of *StTCP23*, which encodes a potato transcription factor derived from an siRNA of PSTVd, could cause PSTVd disease. The authors demonstrated that the expression of *StTCP23* is repressed and accompanied by *StTCP23* transcript cleavage within the identified region of complementarity in PSTVd-infected potato. The authors also claimed that the expression of virulence-modulating-region sequences as 21-nt amiRNAs of *StTCP23* in potato resulted in reduced *StTCP23* transcript abundance and PSTVd-like disease symptoms [41].

The use of RNAi may be an effective approach for engineering crop resistance to a broad range of viroids. Schwind et al. reported that transgenic plants expressing an hpRNA construct derived from PSTVd sequences exhibited resistance to PSTVd infection and that hpRNA-derived siRNAs (hp-siRNAs) could target the mature viroid RNA and prevent its replication [64]. Furthermore, Adkar et al. [65] demonstrated that expressing the viroid–specific small RNAs derived from various functional domains and PSTVd-sRNAs could inhibit PSTVd infection.

Carbonell et al. [66] suggested that amiRNAs and synthetic trans-acting small-interfering RNAs (syn-tasiRNAs) can be used effectively to control PSTVd infections in potato. Small non-coding circular RNAs (250–400 nucleotides) with a compact secondary structure were used to interfere with infections caused by viroids, and the majority of amiRNAs and syn-tasiRNAs were highly active in agroinfiltrated leaves when coexpressed with infectious PSTVd transcripts.

## 7. RNA Silencing-Mediated Multiple Viruses Resistance

In potato, the simultaneous coinfection of multiple viruses may result in pronounced disease phenotypes and lead to crop failure because of synergistic or antagonistic interactions among the viruses. For example, the yield loss can reach 55% under PVX mixed infection with PVY, whereas a single infection with PVX only causes a 15–30% yield loss [67]. An RNA construct carrying two viral RNAs that were cloned as the two arms in a hairpin orientation to generate RNAi against both viruses simultaneously has been reported [68]. Arifa et al. [69] stated that a chimeric expression vector containing three partial gene sequences, derived from the *ORF2* gene of PVX, the *HC-Pro* gene of PVY, and the *CP* gene of PLRV, was constructed to generate transgenic potatoes of cultivars Kuroda and Desiree, with 20% of the transgenic plants being immune to all three viruses because of simultaneous RNA silencing. Other studies have reported a single chimeric hairpin that was expressed in a cassette of fused viral coat protein-coding sequences from PVX, PVY, and PVS as a 600-bp inverted repeat to generate dsRNAs having a hairpin loop configuration. Compared with the severe viral disease symptoms observed in the untransformed controls, the transgenic potato lines (cv. Desiree) showed nearly 100% resistance against PVX, PVY, and PVS infections simultaneously [68].

Moreover, the endogenous tRNA-processing system has been used to generate multiple amiRNAs from tRNA-pre-amiRNA tandem repeats for multiplex gene silencing [70]. An amiRNA-targeting sequence encoding the silencing suppressor HC-Pro of PVY and P25 of PVX was designed by Ai et al. The resulting sequence conferred highly specific resistance against PVY or PVX infection in transgenic potatoes, and this resistance was maintained under conditions of increased viral pressure [71].

In our recent study, the miRNA159 backbone precursor from *Arabidopsis*
*thaliana* was modified to express an amiRNA construct targeting P25, HC-Pro, and Virp1 of PVX, PVY, and PSTVd, respectively. The amiRNA construct was stably transformed in potatoes. Viral challenge assays revealed that these transgenic plants exhibited resistance against PVX, PVY, and PSTVd coinfection simultaneously, whereas the untransformed controls developed severe viral disease symptoms. These results indicated a strategy to engineer resistance to several viruses simultaneously via the c-expression of multiple amiRNAs (unpublished).

## 8. Challenges and Perspectives

RNAi is a direct target of RNA silencing mechanisms that is used to regulate gene expression, control developmental processes, and maintain genome integrity. Moreover, hpRNA-based constructs targeting viral RNAs have proven to be superior to previous transgenic approaches for generating resistance in plants against viruses [72,73]. The major drawback of the topical application of naked dsRNA molecules is their instability in the environment, which is typically reflected in a short antiviral protection time. In addition, RNAi relies on the transformability of a plant when applied against viral infections [74,75].

Viral assembly may be prevented by interfering with the expression of the *CP* gene, which is an important part of the viral structure [76]. As is known, it is extremely easy to cause variation and recombination of viral genomes, and as the CP fragment of the virus was randomly selected to synthesize a dsRNA for transformation, whether the viral resistance of the transgenic plants is exerted against all strains of the virus or against a single strain remains unknown [77]. Therefore, the selection of different CP fragments of viruses to construct dsRNAs may lead to different resistance ranges to those particles.

Selective gene silencing, albeit without absolute specificity, is offered by the RNAi-mediated strategy, and transgene-derived siRNA-based silencing of host genes based on sufficient nucleotide sequence complementarity may represent an off-site targeting risk [78]. Therefore, unexpected phenotypic changes that can significantly impact the agronomic performance of host genes may be caused by this off-target silencing mechanism [79]. Hence, it is necessary to perform bioinformatic analyses of sequence homology to select transgene sequences for minimizing the potential risks of off-target silencing.

amiRNA-mediated gene silencing has a high degree of specificity and effectiveness and enables genome stability and hereditability of phenotypes. Moreover, it has become one of the most important tools in genetic engineering currently. However, different resistance levels were observed when the amiRNAs targeted different viral functional genes or were allocated at different positions in the same functional gene.

When devising strategies for expressing amiRNAs in plants, it may be difficult to stably suppress a target virus gene because of the rapid mutation rate of viruses. Hence, it is essential to target highly conserved regions of the viral genome and express more than one amiRNA sequence in one vector to avoid viral escape resulting from the emergence of evading mutations. It has been proven that the coexpression of multiple antiviral amiRNAs is effective in several viral pathosystems. Despite the successful generation of multiple-virus-resistant plants, immense time would be necessary to generate such amiRNA constructs.

Modern genome-editing techniques, such as the clustered regularly interspaced short palindromic repeats/CRISPR associated 9 (Cas9) (CRISPR/Cas9) and Cas13a/sgRNA (small guide RNA) system, have extended the possibilities of engineering virus resistance in plants [80,81]. Comparing with targeting viral coding sequences like CP of RNAi, the durability of CRISPR-mediated viral targeting can be increased by targeting the viral intergenic region [82]. Moreover, through designing a multiplex CRISPR locus, simultaneous targeting of multiple DNA/RNA viruses could be achieved in plants. High-level PVY resistance was engineered into potato plants by introducing the CRISPR/Cas13a prokaryotic immune system [83]. The excellent potential of the CRISPR technology has rendered it increasingly popular as a tool for the plant in-crop improvement applications of functional genomics [18].

Although the CRISPR technology may represent new technologies to engineer antiviral resistance and in some cases rivals or exceeds the RNAi technologies, it has several disadvantages. Firstly, CRISPR-Cas components can only be carried through transgenic expression, whereas RNAi-based technology is transgene-free; secondly, genome-editing techniques are only feasible within a limited range of potato varieties and it needs a longer time select the edited plants (months or years); thirdly, CRISPR technologies induce permanent gene edits in potato, while RNAi induces transient knockdown of gene expression. Therefore, RNAi approaches remain the main tool for targeting viral pathogens in potato [46,84].

Finally, RNAi will likely continue to be used in potato crop research and development for assessing the function of specific genes because of its high specificity. Providing long-term protection against the targeted viral pathogen to plant cells is a critical future research goal in this field.

## Figures and Tables

**Figure 1 genes-13-00731-f001:**
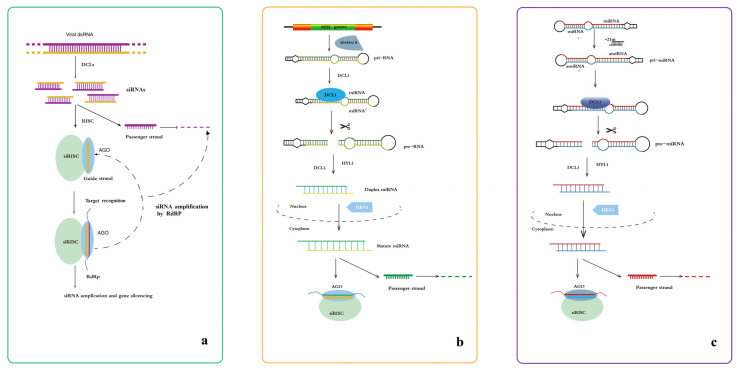
Antiviral RNA silencing pathway in plants. (**a**) The antiviral siRNA pathway. First, the dsRNA structures of viral RNAs are processed by Dicer-like proteins into several siRNA fragments. The guide strand of siRNA (orange) can assemble into functional siRISC, and the passenger strand (purple) is ejected and degraded. All the forms of siRISC contain siRNA bound to an AGO protein. Target RNAs are then recognized by base pairing, and the siRNA populations that engage a target can be amplified by the action of RdRP enzymes, strengthening and perpetuating the silencing response. (**b**) The antiviral miRNA pathway. Primary miRNA (pri-RNA), produced from MIR genes by RNA Pol II, is diced into pre-miRNA by DCL1. Pre-miRNA is further processed by DCL1 and other accessory proteins into duplex miRNA, which is methylated by HEN1 into mature miRNA duplex and transported to the cytoplasm. The guide strand of mature miRNA associates with RISC that consists of AGO1. (**c**) The antiviral amiRNA pathway. The amiRNA transgene expresses a monocistronic MIRNA precursor sequentially processed into an amiRNA targeting a single site in a single viral RNA, and the consequent steps are same as the antiviral miRNA pathway. Several studies have revealed the efficiency of RNA silencing in controlling viral diseases in potato.

**Table 1 genes-13-00731-t001:** Summary of the application of RNA silencing to the viral pathogens of potato.

Virus (Viroid)	Genus	Taget Gene/Protein	RNA Type	Variety	Reference
PVX	*Potexvirus*	PVX-based VIGS vector	siRNA	Desiree	[31]
		VbMS	miRNAs	Katahdin, Russet Burbank	[32]
		P25	amiRNAs	Youjin	Unpublished
PVY	*Potyviridae*	HC-Pro	siRNA	Agria	[33]
		PVY-CP	siRNA	Spunta	[34]
		PVY-CP	siRNA	Atlantic, Ranger Russet	[35]
PVA	*Potyvirus*	PVA-VPg	siRNA	line 2×v2(2-7)	[36]
		PVA-Cp	siRNA	Vales Sovereign	[37]
		PLRY-MP	siRNA	Khufri Ashoka	[38]
PLRY	*Polerovirus*	PLRY-CP	siRNA	Khufri Ashoka	[39]
		PLRY-CP	siRNA	Desiree	[40]
PSTVd	*Pospiviroidae*	*StTCP23*	siRNA	Atlantic	[41]
		*Virp1*	amiRNA	Youjin	Unpublished

## Data Availability

Available data are presented in the manuscript.

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
