# Peer review of "Advances in RNA-Silencing-Related Resistance against Viruses in Potato"

_genes, 2022, doi:10.3390/genes13050731_

Round 1

Reviewer 1 Report

Dear Authors,

it was a great pleasure to read your manuscript.

Please find below my comments and suggestions.

In the paragraph n. 2 where the authors describe the main pathway of the antiviral RNA silencing, the main text is more elaborate than the Figure 1. It is complicated to follow the figure and read the text. The quality of this figure has to be increased to make it more readable. To my opinion the authors should divided the Figure in sections a, b, and c, mention the correct frame in the subparagraph 2.1, 2.2 and 2.3; they should also either comment everything is in the Figure or add more details to the figure as mentioned in the main text.

Probably the text is more readable if the references come right after the authors: for example Line 148 Petrov et al (45)

Line 85: It is not fully clear if the authors are referring to a domain different from PAZ.

Line 108-109: The caption is too thin. More details could be add because the legend is not fully explicative

Line 121-124: please rephrase it because what the authors mention is the opposite of the published article.

Line 122: the authors put a not needed acronym, VSRs

Line 129: is not totally clear who is PVX resistant.

Table 1: some (unpublished) examples are not mentioned in the main text: does it make sense to report them?

Umpublished: to be replaced by unpublished

Line 156-157: it is not fully clear the aim of the sentence. It appears separated from the following phrase.

Line 171: please check if the reference n.51 is correct.

Line 178: Maybe because a typo or a mistake the authors skipped few words from the abstract of the reference crucial to understand the topic

Line 183: I don’t think that the ref 51 is correct.

Line 204: The reference 59 does not seems to be the correct one.

Line 212: Please check the reference 61 as well.

Line 227-229: The title of the reference is particularly explicative. Please think about rephrasing the sentence because it is not fully clear.

Line 401: Faivre-Rampant, O. instead of Faivre, R.O.

Author Response

(1) In the paragraph n. 2 where the authors describe the main pathway of the antiviral RNA silencing, the main text is more elaborate than the Figure 1. It is complicated to follow the figure and read the text. The quality of this figure has to be increased to make it more readable. To my opinion the authors should divided the Figure in sections a, b, and c, mention the correct frame in the subparagraph 2.1, 2.2 and 2.3; they should also either comment everything is in the Figure or add more details to the figure as mentioned in the main text.

Response: Figure 1 has been divided to three sections, and more information was added to be understandable.

(2) Probably the text is more readable if the references come right after the authors: for example, Line 148 Petrov et al (45)

Response: I have changed the place of references and put them after the authors in the whole article.

(3) Line 85: It is not fully clear if the authors are referring to a domain different from PAZ.

Response: "PAZ domain" has been added on L85.

(4) Line 108-109: The caption is too thin. More details could be add because the legend is not fully explicative.

Response: we have added more details follow the title of Figure on L108-122.

(5) Line 121-124: please rephrase it because what the authors mention is the opposite of the published article.

Response: this sentence has been rewritten on L133-136.

(6) Line 122: the authors put a not needed acronym, VSRs

Response: "VSRs" has been deleted.

(7) Line 129: is not totally clear who is PVX resistant.

Response: this sentence has been revised on L141.

(8) Table 1: some (unpublished) examples are not mentioned in the main text: does it make sense to report them?

Response: some examples are not mentioned in the main text have been deleted at table1.

(9) Umpublished: to be replaced by unpublished

Response: "unpublished" has been changed to "Unpublished".

(10) Line 156-157: it is not fully clear the aim of the sentence. It appears separated from the following phrase.

Response: This section has been changed on L170-177.

(11) Line 171: please check if the reference n.51 is correct.

Response: the reference 61 has been changed to a new reference on L462-463.

(12) Line 178: Maybe because a typo or a mistake the authors skipped few words from the abstract of the reference crucial to understand the topic.

Response: this sentence has been revised on L192-193..

(13) Line 183: I don’t think that the ref 51 is correct.

Response: the reference 61 has been changed to a new reference on L439-440.

(14) Line 204: The reference 59 does not seems to be the correct one.

Response: the reference 59 has been deleted and to a reference 59 is used on L213.

(15) Line 212: Please check the reference 61 as well.

Response: the reference 61 has been changed to a new reference on L446-447.

(16) Line 227-229: The title of the reference is particularly explicative. Please think about rephrasing the sentence because it is not fully clear.

Response: This sentence has been changed to a new sentence for understanding clearly on L240-242.

(17) Line 401: Faivre-Rampant, O. instead of Faivre, R.O.

Response: " Faivre, R.O. " has been changed to " Faivre-Rampant, O. " on L413.

Reviewer 2 Report

This is a literature review focused on the use of RNA silencing approaches to engineer antivirus resistance in potato. Particular details are provided for Potato virus X, Potato virus Y, Potato virus A, Potato leaf roll viroid, and Potato spindle tuber viroid.  This review might be of interest to potato pathologists. However, it lacks novelty. The information presented is general to all crop plants and has been extensively reviewed. The most unique part of this document is Table 1. However, Table 1 is not comprehensive.

MAIN CONTRIBUTIONS

  1. The topic is of interest.

POINTS THAT NEED TO BE CLARIFIED

  1. Lack of novelty. The information presented is general, applies to all plants, and has been extensively reviewed. The only unique part of this document is Table 1.
  2. Gene editing is a new technology that is likely to replace RNA silencing approaches in all plants, including potato. This document needs to elaborate on the pros and cons of these competing/alternative technologies as related to potato.
  3. Table 1 has major mistakes on the virus taxonomy and inconsistencies in way target genes are named. The term/pathways dsRNA and hpRNA are not illustrated in figure 1.
  4. The document needs to be edited for appropriate use of the English language.

Author Response

  1. Lack of novelty. The information presented is general, applies to all plants, and has been extensively reviewed. The only unique part of this document is Table 1.

Response: maybe this article is lack of novelty, however, I think it’s necessary to review the latest advances in the development of antiviral strategies to enhance resistance against potato viruses through RNAi. This report would be insightful for the researchers attempting to understand the RNAi-mediated resistance against viruses in potato.

  1. Gene editing is a new technology that is likely to replace RNA silencing approaches in all plants, including potato. This document needs to elaborate on the pros and cons of these competing/alternative technologies as related to potato.

Response: Modern techniques such as genome-editing has been elaborated in the section of Challenges and Perspectives on L317-328.

Table 1 has major mistakes on the virus taxonomy and inconsistencies in way target genes are named. The term/pathways dsRNA and hpRNA are not illustrated in figure 1.

Response: I have corrected the mistake about taxonomy of PVX, and renamed the genes. The term/pathways dsRNA and hpRNA have been changed at Table1.

  1. The document needs to be edited for appropriate use of the English language. Response: I have found native English speaker for linguistic assistance during the preparation of this manuscript.

Round 2

Reviewer 2 Report

This is a revised version. Taxonomic issues in Table 1 were corrected. Mayor points 1 and 2 were not addressed, and continue to be mayor issues with this review:

  1. Lack of novelty. The information presented is general, applies to all plants, and has been extensively reviewed. The only unique part of this document is Table 1.

  1. Gene editing is a new technology that is likely to replace RNA silencing approaches in all plants, including potato. This document needs to elaborate on the pros and cons of these competing/alternative technologies as related to potato.

Author Response

  1. Response: the structure of this manuscript has been modified, which make the text more focused on potato. I think the creative works of this paper are follows: we present an overview of the RNA-silencing-mediated control of plant viral disease in potato and we compare the application of RNAi and gene editing technology in potato.
  2. Response: Although the gene editing may represent new technologies to engineer antiviral resistance and in some cases rivals or exceeds the RNAi technologies, it also has several disadvantages. I have rewritten the pros (L317-327) and cons (L328-335) of gene editing technology in potato.